# RKIP Regulates Differentiation-Related Features in Melanocytic Cells

**DOI:** 10.3390/cancers12061451

**Published:** 2020-06-03

**Authors:** Cristina Penas, Aintzane Apraiz, Iraia Muñoa, Yoana Arroyo-Berdugo, Javier Rasero, Pilar A. Ezkurra, Veronica Velasco, Nerea Subiran, Anja K. Bosserhoff, Santos Alonso, Aintzane Asumendi, Maria D. Boyano

**Affiliations:** 1Department of Cell Biology and Histology, Faculty of Medicine and Nursing, UPV/EHU, 48940 Leioa, Spain; cristina.penas@ehu.eus (C.P.); aintzane.apraiz@ehu.eus (A.A.); yoana.arroyo@kcl.ac.uk (Y.A.-B.); pilarariadna.ezcurra@ehu.eus (P.A.E.); aintzane.asumendi@ehu.eus (A.A.); 2Biocruces Bizkaia Health Research Institute, 48903 Barakaldo, Spain; iraia.munoa@ehu.eus (I.M.); jraserod@andrew.cmu.edu (J.R.); veronica.velascobenito@osakidetza.eus (V.V.); nerea.subiran@ehu.eus (N.S.); 3Department of Physiology, Faculty of Medicine and Nursing, UPV/EHU, 48940 Leioa, Spain; 4Department of Psychology, Carnegie Mellon University, Pittsburg, PA 15213, USA; 5Institute of Biochemistry, Friedrich-Alexander University of Erlangen-Nürnberg, 91054 Erlangen, Germany; anja.bosserhoff@fau.de; 6Comprehensive Cancer Center (CCC) Erlangen-EMN, 91054 Erlangen, Germany; 7Department of Genetics, Physical Anthropology and Animal Physiology, Faculty of Science and Technology, UPV/EHU, 48940 Leioa, Spain; santos.alonso@ehu.eus

**Keywords:** RKIP, melanocytes, melanoma, transcriptome analysis, cell motility, differentiation, biomarker

## Abstract

Raf Kinase Inhibitor Protein (RKIP) has been extensively reported as an inhibitor of key signaling pathways involved in the aggressive tumor phenotype and shows decreased expression in several types of cancers. However, little is known about RKIP in melanoma or regarding its function in normal cells. We examined the role of RKIP in both primary melanocytes and malignant melanoma cells and evaluated its diagnostic and prognostic value. IHC analysis revealed a significantly higher expression of RKIP in nevi compared with early-stage (stage I–II, AJCC 8th) melanoma biopsies. Proliferation, wound healing, and collagen-coated transwell assays uncovered the implication of RKIP on the motility but not on the proliferative capacity of melanoma cells as RKIP protein levels were inversely correlated with the migration capacity of both primary and metastatic melanoma cells but did not alter other parameters. As shown by RNA sequencing, endogenous RKIP knockdown in primary melanocytes triggered the deregulation of cellular differentiation-related processes, including genes (i.e., ZEB1, THY-1) closely related to the EMT. Interestingly, NANOG was identified as a putative transcriptional regulator of many of the deregulated genes, and RKIP was able to decrease the activation of the NANOG promoter. As a whole, our data support the utility of RKIP as a diagnostic marker for early-stage melanomas. In addition, these findings indicate its participation in the maintenance of a differentiated state of melanocytic cells by modulating genes intimately linked to the cellular motility and explain the progressive decrease of RKIP often described in tumors.

## 1. Introduction

Raf Kinase Inhibitor Protein (RKIP), also known as phosphatidylethanolamine-binding protein 1 (PEBP1), was originally described as an inhibitor of the MAPK and ERK1/2 pathway [1] although latter studies revealed its additional role as a regulator of other signaling cascades including the G-protein-coupled receptor (GPCR), glycogen synthase kinase 3β (GSK3β) and nuclear factor kappa B (NF-κB)-driven ones [2]. As a signaling switch, RKIP plays a role in essential processes such as differentiation, proliferation, and cell survival. Moreover, deregulation of RKIP expression has been associated with a wide variety of disorders such as neurological diseases, diabetes, altered spermiogenesis and cancer [3,4,5].

Reduced or lack of RKIP expression level has been proposed as a marker of poor prognosis in multiple types of cancer including esophageal, gastric, breast, prostate, pancreatic, cervical cancers and gliomas [6,7,8,9,10,11]. Regarding melanoma, published data report the role of RKIP regulating cell proliferation, migration and invasion capability [12,13,14], by mechanisms leading to NFκB and Ras-ERK1/2 pathway inhibition. Nevertheless, there is little known on the pathways regulated by RKIP in normal melanocytes nor on its role in the malignant transformation of this type of cells. Some studies [14,15] describe a malignancy-related gradual reduction of RKIP levels in melanoma patients, but however, these were performed with a small cohort of patients; a larger study would be required in order to establish the real diagnostic or prognostic value of this marker in melanoma.

Malignant transformation of cells implies the acquisition of particular characteristics in terms of survival and proliferative capability, while metastatic spreading depends on enhanced migration and invasion [16]. In addition, tumor persistence seems to rely on a particular subset of cells with acquired “stem cell-like” properties [17,18]. Similar to embryonic stem cells, this subpopulation of tumor stem-like cells is able to grow intensively and to infiltrate local tissue [11,19]. The self-renewal capability of embryonic stem cells is regulated by pluripotency-related transcription factor such as Nanog homeobox (*NANOG*), POU class 5 homeobox 1 (*OCT4*) and SRY-box transcription factor 2 (*SOX2*) [20] which are also aberrantly expressed in many malignant human tumors [21,22,23]. In melanoma, melanosphere formation has been found to increase NANOG expression [24]. Moreover, NANOG and OCT4 overexpression increases motility and transmigration of melanoma cells [25]. In addition to the well described role of NANOG as a crucial regulator of stemness maintenance, this transcription factor has been also implicated in the regulation of the epithelial-mesenchymal transition (EMT) [26]. The EMT process has been classically linked to an increased motility of cancer cells favoring dissemination of the disease [27]. This process, characterized by a cellular reprogramming towards a more mesenchymal phenotype, is often driven by the Snail transcriptional repressor (SNAIL), the bHLH transcription factor TWIST, and the Zinc finger E-box-binding homeobox (ZEB) families [27]. Of note, NANOG has been described as a direct activator of EMT-associated genes including *ZEB1*, *ZEB2*, and microRNA-21 (*miR-21*) [28]. Interestingly, the literature describes an intertwisted signaling system between SNAIL and RKIP; SNAIL has been shown to repress transcription of RKIP while simultaneously SNAIL has also described as a downstream target of RKIP [29]. Furthermore, a negative correlation has been found among RKIP and expression of EMT regulators including ZEB1, ZEB2, and SNAIL [29] although the underlying mechanism remains elusive.

The generalized loss of RKIP expression in tumor tissues together with its described capability to modulate pathways driving tumor cell proliferation or an invasive phenotype, warrant a more profound analysis of RKIP-dependent pathways on primary cells in order to determine profile alterations that may influence cellular transformation or promote an aggressive phenotype. Thus, in this work, we examined the role of RKIP in the biology of primary melanocytes and malignant melanoma cells. In addition, we evaluated the diagnostic and prognostic value of RKIP with a special emphasis on the early stages of the disease. Our results suggest that RKIP is implicated in the maintenance of a differentiated state of melanocytic cells and support its utility as a diagnostic marker for the evaluation of early-stage melanomas.

## 2. Results

### 2.1. Immunohistochemical Characterization of RKIP Protein Expression in Bening Nevi and Melanoma Biopsies 

The pattern of RKIP protein expression in human melanocytic lesions was determined by immunohistochemistry (IHC) and included 239 melanoma and 75 nevi samples. Representative IHC images of most common lesions (compound nevus, superficial spreading melanoma, and nodular melanoma) are shown in Figure 1a, while the statistical analyses are summarized in Figure 1b–e. Nevi samples exhibited higher positivity for RKIP staining compared with the whole cohort of melanoma samples (Figure 1b); 94% of nevi samples were positive for RKIP whereas only 51% of melanoma cases presented positive staining. Interestingly enough, in situ melanomas, characterized by an excellent prognosis upon surgical removal, exhibited a strong positive RKIP expression in almost 80% of cases (Appendix A). Of note, RKIP staining displayed minimum intrasample variation and a cytoplasmic localization. Univariate analysis confirmed the statistical significance of observed differences among nevi and the entire set of melanoma samples (q < 0.001) (Figure 1b) and both, univariate and multivariate analysis, provided statistical evidence for a different level of expression of RKIP in nevi and melanomas at early stages (AJCC 8th I/II) (Figure 1c). Moreover, by means of a logistic regression analysis (Nevus = 0, Melanoma = 1) to control for age and sex as covariates (Figure 1c), a polynomial contrast expansion in RKIP demonstrates that this association is linear (β = −2.288, q < 0.001), i.e., linear increments in protein levels correlate significantly with a larger probability of the biopsies of being identified as nevus, while quadratic effects tend to be moderate and non-significant (β = 0.465, q = 0.218). On the other hand, changes in the levels of RKIP did not statistically associate with metastasis development either when including the whole spectrum of melanoma staging (q = 0.132), or if we only consider the clinical-risk subset of melanoma subjects (AJCC I + II) (q = 0.499) (Figure 1d). Likewise, no significant pattern-differences of this protein were observed between AJCC stages I and II (q = 0.520). Nonetheless, we observed an association among RKIP staining and Breslow thickness as melanoma biopsies with higher RKIP protein level tended to display significantly lower values of Breslow thickness (q = 0.014) across all AJCC stages (Figure 1e). In summary, clear RKIP staining differences were observed among benign (nevi) and early stage (I–II) melanomas. In addition, RKIP staining could not predict disease progression but high RKIP level correlated with lower Breslow thickness in samples of all melanoma stages.

### 2.2. Determination of RKIP Expression on Melanocytes and Melanoma Cell Lines

Melanocytes and several malignant melanoma cell lines were selected in order to further decipher the mechanisms underlying observed differences in human samples. Prior to molecular and cellular behavior analysis, we evaluated the expression of RKIP (both mRNA and protein) in three primary melanocyte cells and eleven melanoma cell lines by RT-qPCR and Western Blot (Figure 2). The relative quantification of the mRNA levels evidenced a generalized reduction of *RKIP* expression in melanoma cell lines (*p*-value = 0.0001) compared to primary normal melanocytes (Figure 2a). Of note, similar mRNA levels were detected when comparing primary (light green) and metastatic (light blue) melanoma cell lines. Analysis of RKIP protein levels (Figure 2b, details of Western Blot in Appendix A) revealed a consistent reduction in melanoma cell lines with no differences among primary (Mel-HO, A375) and metastatic (HT-144, Hs-294T, Colo-800) cell lines. Results obtained from analyzed cell lines recapitulated those obtained on human biopsies. In addition, observed correlation between the RKIP mRNA expression and RKIP protein-content suggested a deregulation at the transcriptional level (Figure 2b). 

### 2.3. Involvement of RKIP on Malignancy-Related Properties of Melanoma Cells

Gradual loss of RKIP expression has been linked to tumor malignancy [9] and RKIP has been often implicated on tumorigenic properties of cancer cells such as altered proliferation and enhanced migration capability. To study the involvement of RKIP in the pathogenesis of melanoma, we modified the endogenous RKIP levels. Downregulation of endogenous *RKIP* was accomplished by RKIP shRNA lentiviral particles, while RKIP-overexpressing plasmids were used to increase cellular RKIP levels.

Downregulation by shRNA led to a decrease of up to the 70–80% on the endogenous RKIP mRNA level on selected primary melanoma cell lines (Figure 3a) which was also consistent with a reduction on the protein percentage (Figure 3b, details of Western Blot in Appendix A). Reduction of endogenous RKIP by lentiviral silencing did not alter proliferation capability of A375 and MelHO cells (Figure 3c). By contrast, the RKIP-downregulated primary melanoma cells showed a significant increase in motility, assessed both by wound healing and collagen-coated transwell assays (Figure 3d,e).

To reinforce our data regarding the involvement of RKIP in melanoma cell motility, MeWO and A2058 metastatic melanoma cell lines were transfected with a RKIP-overexpressing plasmid resulting in a 5- and 15-fold increase of RKIP-mRNA level, respectively (Figure 4a). In addition, we detected a concomitant elevation of intracellular RKIP-protein percentage (Figure 4b, details of Western Blot in Appendix A). Consistent with our previous results in primary melanomas, the increase in cellular RKIP level led to a decrease in the migration capability of melanoma cells (Figure 4c). Surprisingly, both analyzed cell lines showed different behavior on the active migration assay (Figure 4d); thus, while no differences were observed in MeWO cells, RKIP overexpression clearly diminished the capacity of A2058 cells to pass through a collagen-based barrier. Of note, basal collagen-through migration activity of MeWO cells was significantly lower than that of A2058 cells. Briefly, cellular RKIP levels were inversely correlated with the migration capability of both, primary and metastatic melanoma cell lines, while no major effect was detected on cellular proliferation.

### 2.4. Transcriptome Modulation by RKIP Downregulation in HEMn-LP Cells

With the aim of elucidating the molecular mechanisms whereby RKIP could modulate processes related to cellular malignancy, *RKIP* was downregulated in primary melanocytes (HEMn-LP) by the above described shRNA lentiviral particles. Infection resulted in a 70–80% reduction of RKIP mRNA and 40% of protein level (Figure 5a, details of Western Blot in Appendix A).

Two independent replicates of control (shCTR) and RKIP knockdown (shRKIP) HEMn-LP samples were subjected to RNA sequencing. The first part of the analysis focused on the identification of a set of differentially expressed genes between shCTR and shRKIP HEMn-LP based on standard threshold Log2FC ≥ 1, *p*-value ≤ 0.05 and False Discovery Rate (FDR) ≤ 0.05. The resulting 224 differentially expressed genes were used for monitoring the functions and pathways that were mainly affected in melanocytes due to the decreased RKIP expression. Expression patterns of altered genes are shown in Figure 5b. The set of genes with modified expression were roughly equally divided into over- (113) and under-expressed genes (111). The Log2FC, *p*-values, and FDR for each gene are detailed in Appendix A.

In order to gain insight into the functional characteristics of detected changes, over- and under-expressed genes after RKIP silencing were subjected to Gene Ontology (GO) (Figure 5c) and pathway (Kyoto Encyclopedia of Genes and Genomes, KEGG) (Appendix A) enrichment analyses. RKIP knockdown on HEMn-LP cells displayed a transcriptional misregulation in the GO term ‘cancer gene signature’ (*p*-value < 0.001; Appendix A). Moreover, the set of genes with altered expression upon endogenous RKIP reduction showed an enrichment in a variety of essential processes including developmental pigmentation, proliferation and developmental and cell differentiation (*p*-value < 0.05; Figure 5c). Interestingly, RKIP knockdown led to the downregulation of essential melanocyte-pigmentation genes such as *PMEL* (Melanocytic linage-specific antigen, 2-fold decrease, *p* value 0.0003, FDR 0.04), *MLANA* (Melanoma Antigen recognized by T-cells, 8-fold decrease, *p* value 0.001, FDR 0.02), *GPR143* (G-Protein Coupled Receptor 143,11-fold decrease, *p* value 10^−5^, FDR 0.01) and *TYRP1* (Tyrosinase-related protein 1, 5-fold decrease, *p* value 10^−6^, FDR 0.007). On the other hand, only *KIT* (proto-oncogene KIT,) was upregulated among the deregulated genes belonging to the developmental pigmentation group (2.3-fold increase, *p* value 0.0001, FDR 0.02).

Development and differentiation showed the best FDR value among significantly enriched biological processes. This signature encompassed 83 genes which represented 37% of the total altered gene-set and included HOX family members, proto-oncogene *KIT*, proto-oncogene *MYC*, *ZEB1*, and Thy-1 cell surface antigen (*THY-1*), among others (Appendix A). We focused on genes belonging to this process due to the statistical robustness of this group on our data set as well as to the intimate link among this particular process and the cellular migration-capability. As shown in Figure 5d, downregulation of endogenous RKIP led to an increase in the expression of selected genes, validating the RNA Seq data (Appendix A).

Interestingly, neurotrophic receptor tyrosine kinase 2 (*NTRK2*), *ZEB1*, and *THY-1* are not only implicated in developmental processes, as they also known regulators of cellular migration. Based on our previous results that implicated RKIP on the migration capability of melanoma cells, we made use of RKIP overexpression to analyze the effect on *ZEB1*, *NTRK2*, and *THY-1* transcription. RKIP-driven transcriptional repression was confirmed by RT-qPCR for *ZEB1* and *THY-1* in both cell lines (A2058 and MeWO) while *NTRK2* revealed a cell type-dependent response (Figure 5e) Taking together, RKIP revealed the capacity to modulate genes involved in essential processes (e.g., Development and differentiation) and to repress genes with described roles in cellular migration.

### 2.5. NANOG as a Putative Transcription Factor Regulated by RKIP

RKIP has no described function as a direct transcriptional regulator. Thus, observed transcriptional alterations imply the presence of yet unknown transcription factors or regulators downstream RKIP. We focused on deregulated genes belonging to Development and differentiation and conducted an in silico approach in order to detect potential transcription factors acting between RKIP and its downstream modulated genes. Seventy-one percent of genes in this category were putative targets of NANOG transcription factor (Figure 6a, Appendix A). NANOG is a transcription factor involved in the maintenance of stemness and often linked to cancer aggressiveness. Therefore, we wonder whether RKIP could somehow modulate NANOG expression. To analyze this point, we made use of a construct encoding the NANOG promoter attached to the Enhanced Green Fluorescent Protein (EGFP) coding sequence, and cells were co-transfected with either empty plasmid (pCTR) or RKIP-coding plasmid (pRKIP). Activation of NANOG promoter was determined as the percentage of cells expressing EGFP. As shown in Figure 6b, increased RKIP expression led to a significant decrease on NANOG promoter activation; a similar effect was observed in both cell lines.

The gene miR-21 is a described target for NANOG [28]. To further validate the implication of RKIP in *NANOG* regulation, we determined miR-21 transcription level upon RKIP overexpression. As shown in Figure 6c, the expression of miR-21 was significantly lower on RKIP-overexpressing cells. These findings point towards the involvement of NANOG downstream RKIP in the regulation of gene expression.

## 3. Discussion

The molecular alterations involved in the pathogenesis of melanoma represent a topic of active research, which has enabled the identification of disease-associated key, oncogenes, and tumor suppressor genes providing scientific foundation for urgently needed therapeutic approaches [30,31,32,33]. In addition, the high tendency of melanoma to metastasize makes it also important to improve on diagnostic and prognostic markers for early melanoma in order to better discriminate among patients with low and high risk for metastasis.

Several studies have shown that RKIP exhibits low expression levels in various cancers and it is often absent in metastases [5,8,9,10,34,35,36,37,38,39,40,41,42,43,44,45,46,47,48]. In agreement, decreased RKIP expression has been associated with metastatic uveal melanoma while low levels of RKIP were detected on both metastatic as well as non-metastatic cutaneous melanoma biopsies [15,49]. These studies, although interesting, were carried out with small cohorts of patients. In addition, in those studies claiming the association among low RKIP expression and metastasis, decreased RKIP expression was assessed by comparison of primary tumors and biopsies at metastatic sites [8,9,15]. Results obtained from the aforementioned works revealed a clear malignancy-related silencing of RKIP on tumor cells, although they did not analyze the possible predictive role of RKIP. Our study, including 75 nevi and 239 samples of malignant melanoma, allowed deepening on the diagnostic and prognostic value of this ubiquitous biomarker. Of note, all melanoma biopsies were obtained from the primary lesion, which may explain the lack of statistical RKIP-staining differences among stage I–IV melanomas. Also, the cohort size of stage III and IV melanoma patients was small (when comparing to stage I–II patients) and we cannot discharge its effect when analyzing all stages together.

Furthermore, we also focused on early stage (stage I–II according to AJCC 8th classification) melanomas in order to evaluate its usefulness to discriminate among patients with good and bad evolution of the disease. Here, our results agree with previous studies on the diagnostic capability of RKIP staining, as melanoma samples exhibited an overall decrease in staining when compared with benign lesions (i.e., nevi). Unfortunately, RKIP staining was not able to distinguish stage I–II patients with a favorable evolution of the disease from those who eventually developed metastasis. Nevertheless, it is worth mentioning the association among strong RKIP staining and lower Breslow index across all melanoma stages (stage I–IV) suggesting that RKIP may not determine tumor malignancy but may be related to the primary tumor position or progress through the skin.

Classically, IHC-mediated routine identification of melanocytic lesions include the use of melanocyte and melanoma markers, like tyrosinase (TYR) and tyrosinase-related proteins (TYRP1 and DCT), gp100 and Melan-A [50]; nonetheless, the utility of a combined immunohistochemical analysis including Bcl-2, nuclear S100A4, Ki67 and MITF to improve the risk stratification of early-stage malignant melanoma patients has been recently reported [36]. In our study, we did not find a relationship between RKIP expression and metastatic progression of melanoma. However, we think that additional studies are needed to further explore if the combination of strong RKIP expression and low Breslow index should have better predictive value than any of these markers individually.

Delving into the study of RKIP protein as regards the pathogenesis of melanoma, we carried out molecular and functional assays using melanocytes and melanoma cell lines. In accordance with our histopathological results and previously published studies [14,36], we found that both RKIP mRNA and protein expression were significantly lower in melanoma cell lines than in primary cultures of melanocytes with the exception of the Mel-HO cell line; this cell line exhibited RKIP mRNA level similar to that observed on melanocytes but a reduced protein content that suggests the involvement of a post-transcriptional mechanism limiting translation. Of note, RKIP has been described as a target for several microRNAs able to regulate cellular protein level [2]

RKIP is a multifunctional protein involved in carcinogenesis regulating cellular growth, motility, epithelial to mesenchymal transition and invasion [51,52,53]. Several authors have suggested that RKIP may not have a significant role in primary tumors but that instead, this protein could play an important role as a metastatic suppressor [5,8,9,10,34,35,36,37,38,39,40,41,42,43,44,45,46,47,48]. In this sense, and despite the described role for RKIP in the regulation of the MAPK/ERK pathway, RKIP has been implicated on the invasive behavior of malignant melanoma cells but not on their proliferative capability [14]. Moreover, Schoentgen and Jonic [16] described the involvement of RKIP on the cortical actin organization during the membrane changes that happen during tumor cell migration. In agreement with previous studies, we confirmed the implication of RKIP on the motility of malignant melanoma cells as RKIP expression was inversely correlated with the migration capability of both, primary and metastatic melanoma cell lines. Nevertheless, modulation of the cellular RKIP level did not show an influence on the proliferative activity of melanoma cells. Therefore, considering the relevance of cellular motility on tumor metastasis, these results support a role for RKIP loss in melanoma dissemination.

To define the cellular mechanisms regulated by RKIP that could explain the selective force favoring a decreased presence of this protein on melanomas when comparing with benign lesions (i.e., nevi), RKIP gene was silenced using lentivirus in primary melanocytes and RNA sequencing were performed to analyze the transcriptome changes derived from RKIP modulation. The transcriptome of melanocytes after RKIP silencing revealed a transcriptional misregulation in cancer gene signature. Among others, this signature included altered expression pattern of the oncogenes *KIT*, *BCL3*, *MAF*, *MYC*, *MYCL*, *HOXA9*, *CDC25B*, and *PIM1*. In our data, all of them showed a two to five-fold increase, supporting the role for RKIP like a tumor suppressor gene [54]. Interestingly enough, downregulation of RKIP expression on melanocytes resulted in the alteration of cellular processes intimately linked to malignant transformation of cells, such as development and differentiation. Moreover, developmental pigmentation, a process specifically linked to the melanocytic lineage, was also enriched. According to our RNA-seq data, RKIP would be a repressor of *KIT* and an inducer of *TYRP1*, *MLANA*, and *PMEL* gene expression, among others. *TRYP1*, *MLANA*, and *PMEL* are among the best-known transcriptional targets of the master melanogenic regulator microphthalmia-associated transcription factor (*MITF*) [55] and it would be of interest to further analyze the possible crosstalk among RKIP and MITF. In addition, data indicate that RKIP represents a brake for the EMT process, by regulating the expression of genes such us *ZEB1*, *THY*-1 and *NTRK2.*

Scientific evidence demonstrates that in a heterogeneous tumor mass, those cells responsible for drug-resistance, recurrence and metastasis contain characteristics of stem cells, that is, the ability to self-renew and differentiate in any cell type of the tumor mass [56]. In this work, after silencing of RKIP in HEMn-LP melanocytes, more than 70% of the differential expression genes belonging to development and differentiation were found to be putative targets of NANOG. NANOG has been identified as one of the crucial inducers of this stem cell-like state type [20] and is aberrantly expressed in many kinds of tumors [21,22,23]. We observed that transient forced-increase of RKIP expression in metastatic melanoma cells led to the decrease of *NANOG* promoter activation pointing towards a functional relationship among RKIP and *NANOG* expression. In line with these results, Lee et al. [57] noticed a high amount of crosstalks between pathways regulated by RKIP and those under the control of main stemness transcription factors (i.e., *OCT4*, *KLF4*, *SOX2*, and *NANOG*) and proposed RKIP as a regulator of the differentiation state of cells. This hypothesis would be in agreement with the stronger RKIP expression found in differentiated melanocytes from nevi lesions when comparing with melanoma samples.

In addition to the maintenance of the stemness, NANOG has been also implicated in the EMT [19,26] and by regulating the expression of *ZEB1* and *THY-1* among other genes [28]. In fact, EMT and development of stemness properties are often closely related processes [58]. As previously mentioned, these two genes are among those with deregulated expression in our RNA-seq study. ZEB1 is one of the major activators of the EMT program and increasing evidence places ZEB1 also as an important regulator of differentiation, proliferation, DNA damage response and cell survival [59]. Interestingly, ZEB1 is among the transcription factors driving the early hybrid EMT state and hybrid EMT states (i.e., states with intermediate characteristics among fully epithelial and fully mesenchymal cells) have been linked to collective cells migration and highest metastatic potential [58]. This result, together with the observed modulation of the cellular migration capacity driven by RKIP, are in line with the rapid RKIP diminution observed on malignant lesions, even at early stages, as well as the association among low Breslow index and presence of RKIP. In fact, capacity of a tumor to deepen on the skin requires the acquisition of characteristics as those blocked by RKIP. On the other hand, THY-1 is a protein implicated in the endothelium transvasation of melanoma cells during metastasis spreading [60]. These results could be indicating the implication of RKIP loss in the plasticity required for the intra- and extravasation during melanoma metastasis. In this context, we have also found that the expression of miR-21 was significantly lower on RKIP-overexpressing cells. miR-21 is a known target for NANOG [28] and an important inducer of EMT affecting migration and invasion capability [61,62,63] suggesting a possible role for this onco-miRNA in melanoma malignancy [28,64]. These results point towards the involvement of NANOG downstream RKIP in the regulation of gene expression related to malignant phenotype of melanoma cells.

To summarize, our study supports the diagnostic utility of RKIP staining due to the significantly lower RKIP protein levels in melanoma samples, even at early stages (I–II) of the disease. The capability of RKIP to downregulate the transcription of genes involved in motility-related processes such as EMT as well as the migration process itself, may partially explain its generalized negative selective-pressure on tumors. Additionally, we propose that RKIP could play a role in the maintenance of the differentiation state by negatively regulating *NANOG* gene expression although further research would be required for a better description of the underlying mechanism.

## 4. Materials and Methods

### 4.1. Patients

A total of 314 patients (239 melanomas and 75 nevi) were recruited at the Dermatology Units at the Basurto and Cruces University Hospitals between 1990 and 2016 (Table 1). Inclusion criteria were: (1) a histologically confirmed diagnosis of nevus or malignant melanoma; (2) no treatment except primary surgery; (3) no infection as judged by clinical evaluation and the absence of increased infectious parameters in the blood. After surgery of the primary tumor, clinical check-ups were scheduled every three months for the first two years of the follow-up, and every six months thereafter, until a five-year follow-up had been completed. Annual revisions were then scheduled up to the tenth year post-surgery. The patients who developed metastasis during the follow-up period were again examined every three months for two years after metastasis had been diagnosed. The presence or absence of metastasis was assessed in all patients by physical examination, as well as through laboratory and radiological testing (X-rays and/or computed tomography –CT- scanning). Metastases were detected in 92 of the 239 melanoma patients (38%) during this study, including those for whom the disease had spread at the moment of diagnosis. Disease stages were classified according to the AJCC 8th edition. The clinical and diagnostic data for each patient were collected retrospectively from centralized electronic and/or hard-copy medical records. For the statistical prediction analysis, only melanoma patients at early disease stages (I and II) were included, and the inclusion of the “disease-free” group required a minimum tracking of 2 years.

The study was conducted in accordance with the Declaration of Helsinki principles and it was approved by the Euskadi Ethics Committee (reference 16-99). Written informed consents were obtained from all the subjects. The melanoma biopsies collected were stored at the Basque Biobank until use (https://www.biobancovasco.org/).

### 4.2. Immunohistochemistry and Statistical Analysis

Five μm-thick sections of FFPE blocks were analyzed by immunohistochemistry of RKIP (anti-PBP Antibody, ab76582, ABCAM, Cambridge, UK) (Kit EnvisionTMG|2 Sistema/AP, Dako Corporation, Denmark; 3 μg/µL; antigen retrieval: citrate buffer low pH 6.1–steam for 105 min). Slides were finally counterstained with hematoxylin and the pictures were taken using a slide scanner NanoZoomer S210 Digital slide scanner (Hamamatsu C13239-01) (PiE-UPV/EHU). Analysis of samples was conducted following a manual semi-quantitative method; categories were determined based on staining intensities (negative (0), weak (1) and strong (2)) as shown in Appendix A. Samples displayed a uniform intra-specimen staining and RKIP marker presented high affinity to melanocytic and melanoma cells (no staining observed in surrounding cells). Due to the color similarities among melanin, macrophage pigment, and chromogen signal, any misinterpretation was avoided by the analysis of a sequential slide stained with hematoxylin-eosin. The staining intensity was evaluated as negative (0), weak (1), and strong (2). The specimens were independently evaluated by two observers and discordant cases were jointly reviewed followed by a conclusive judgment. Data were analyzed using R 3.4.4. Univariate statistical testing between categorical features (Sex, Melanoma Evolution and Diagnosis between Melanoma and Nevus) was assessed by a Pearson’s chi-squared test, whereas the Cochran-Armitage Test was employed when ordinal variables were involved (RKIP and AJCC stage). Likewise, statistical differences between two or more groups were estimated by a two-tailed Kruskal–Wallis test or ANOVA, if normality assumption as measured by the Shapiro–Wilk test is satisfied. Finally, a general lineal model was employed for multivariate analysis, with an orthogonal polynomial contrast expansion in RKIP, assuming equally spaced levels. All *p*-values were computed non-parametrically using the R package “coin” [65] and corrected for multiple comparisons using an FDR controlling procedure [66].

### 4.3. Cell Lines and Proliferation, Migration and Invasion Assays

In the present work, eleven melanoma cell lines and three primary melanocytes were studied. The primary human melanocytes were purchased from Invitrogen, Carlsbad, CA, USA (#C-002-5C for lightly pigmented neonatal foreskin, HEMn-LP; #C-102-5C for moderately pigmented neonatal foreskin, HEMn-MP; and #C-202-5C for darkly pigmented adult foreskin, HEMn-DP). All primary human melanocytes were grown in Cascade Medium 254 supplemented with Cascade Human Melanocyte Growth Supplement (both from Invitrogen; Carlsbad, CA, USA) in the absence of antibiotics.

Likewise, we cultivated eleven different melanoma cell lines: A375(ATCC CRL-1619), A2058 (ATCC CRL-11147), Hs294T (ATCC HTB-140), HT-144 (ATCC HTB-63), MeWo (ATCC HTB-65), WM793B (ATCC CRL-2806), and 1205Lu (ATCC CRL-2812) were purchased from American Type Culture Collection (Rockville, MD, USA); as RPMI7951 (ACC76), COLO-800 (ACC193), MEL-HO (ACC62), and MEL-Juso (ACC74) were obtained from Innoprot (Derio, Bizkaia, Spain). The melanoma cell lines were cultured in appropriate medium supplemented with 10% fetal bovine serum (FBS), 2 mM L-glutamine and antibiotics according to the manufacturer’s instruction. All primary human melanocytes and melanoma cell lines were cultured at 37 °C with 5% CO_2_ and 95% humidity.

For the proliferation capacity, the control melanoma cells (with an empty vector) and stable RKIP transfected clones were subjected to 2,3-bis-(2-methoxy-4-nitro-5-sulfophenyl)-2H-tetrazolium-5-carboxanilide (XTT) assay, following the manufacturer’s instructions. The readings at 24, 48, and 72 h were normalized, and percent surviving cells were calculated against control transfected cells. In parallel, Western Blotting analysis was performed for checking the transfection.

For migration capacity assays, after seed the cells in 24-well plate, the monolayers were incubated with 0.5 µg/mL of Mitomicyn C during 2 h and then, were scraped with sterile plastic micropipette tip. The wound closure was photographed microscopically at 0, 24, and 48 h.

The transwell active migration assay was performed using Type I-Collagen coated inserts with 6.5-mm-diameter polycarbonate filters (8-μm pore size). Cells (1 × 10^4^) suspended in 200 μL of DMEM without FBS, were seeded in the top chambers. The bottom chambers were filled with 300 μL of DMEM containing 10% FBS (as a chemoattractant). Cells were allowed to migrate overnight. The non-migrated cells on the upper surface of the filter were carefully and thoroughly were removed with cotton swabs. Migrated cells were fixed with a mix of cold 4% paraformaldehyde plus 2% ethanol and stained with crystal violet. Five images per insert were taken using a compound optical microscope and migrated cells were quantified by ImageJ Software. Results were expressed as the average number of migrated cells per well obtained from three separate experiments done in triplicate.

### 4.4. Transfection Assays

Normal melanocyte cell line HEMn-LP and A375 and MelHO primary melanomas were transduced with RKIP shRNA Lentiviral Particles (sc-36430-V, 2MOIor Control shRNA Lentiviral Particles (sc-108080 2MOI) (Santa Cruz Biotechnology Inc., Dallas, TX, USA) following the manufacturer´s instruction. Two days after infection, the cells were selected with 5 µg/mL of Puromycin (P8833, Sigma Aldrich, San Luis, MO, USA,) to get stable purified cell lines.

A2058 and MeWO metastatic melanoma cell lines were transfected with overexpressing plasmid for RKIP (RC206355, OriGene Technologies, Inc, Rockville, MD, USA) or empty plasmid using Lipofectamine 2000 (Thermo Fisher Scientific, Waltham, MA, USA) according to the manufacturer’s instructions. All of the transfection experiments were performed with 500 ng of DNA. The experimental assays were performed at least after 24 h of transfection.

To study the activity of NANOG induced by RKIP, NANOG promoter was cloned into a plasmid vector containing the coding region of the enhanced green fluorescence protein (EGFP) gene (PL-SIN-Nanog-EGFP #21321, Addgene, Watertown, MA, USA). Co-transfection of this plasmid and RKIP overexpressing plasmid was performed in metastatic melanoma cell lines A2058 and MeWO. After 48 h post-transfection, the positive cells for fluorescence were determined by means of a Zeiss Fluorescence Microscope.

### 4.5. Western Blotting Analysis

Melanoma cells and primary melanocytes were harvested by trypsinization, washed with PBS and lysed in RIPA lysis buffer (80 mM Tris-HCl pH 8, 150 mM NaCl, 1% NP 40, 0.5% sodium deoxycholate, 0.1% SDS) containing Protease Inhibitor Cocktail (Sigma-Aldrich Quimica, S.A., Madrid, Spain) for 15 min on ice. Lysates were then cleared by centrifugation at 10,000 g for 5 min. Total protein concentration was determined using the bicinchoninic acid assay.

For RKIP protein detection, 40 μg of total proteins from each sample were resolved by electrophoresis on SDS-polyacrylamide gel and then transferred onto a nitrocellulose membrane (Whatman GmbH, Dassel, Germany). The blots were incubated with PBS containing 5% Bovine Serum Albumin and 0.1% Tween-20 for 1 h to block non-specific binding, and then incubated with an appropriate dilution of primary antibody at 4 °C for overnight (PEBP1 #372100, 1:5000, Life Technology, Carlsbad, CA, USA; γTubulin #T9026, 1:3000, Sigma-Aldrich, San Luis, MO, USA). The membrane was washed with TBST for three times (10 min/each time), then incubate with goat anti-mouse Horseradish Peroxidase (HRP) conjugated secondary antibody (anti-mouse #1032-05, 1:10,000, Southern Biotechnology, Birmingham, AL, USA) for 2 h at room temperature. Finally, proteins were visualized by enhanced chemiluminescence using the SuperSignal^®^ West Pico Chemiluminescent Substrate (Thermo Scientific, Rockford, IL, USA).

### 4.6. RNA Extraction and Quantitative Real-Time Polymerase Chain Reaction

Total RNA from cultured cells was isolated using RNeasy Mini kit (Qiagen Inc, Hilden, Germany) and for each sample, cDNA was synthesized from 1 µg total RNA using the iScriptTM cDNA Synthesis kit (Bio-Rad, Hercules, CA, USA) according to the manufacturer’s instruction. Quantitative real-time PCR (RT-qPCR) assays were carried out using an iCycler PCR platform (Bio-Rad, Hercules, CA, USA). The reaction mixture contained 0.1 µL cDNA from the reverse transcription reaction, together with forward and reverse specific primers and iQTM SYBR^®^ Green Supermix (Bio-Rad, Hercules, CA, USA) in a final reaction volume of 20 µL. 

The sequences of primers used were:*RKIP*-Fw:5′-AATAGACCCACCAGCATTTCG-3′/Rev:5′-TGCCACTGCTGATGTCATTG-3′. *ZEB1*-Fw:5′-GTGCAGTTACACCTTTGCA-3′/Rev:5′-CACATGTCTTTGATCTCTTCCT-3′. *NTRK2*-Fw:5′-CTCCCGGAATTGGGTTGGAG-3′/Rev:5′-GGGGCGCAGATTCCTTGTTA-3′. *THY-1*-Fw:5′-GTTTGACCAGGAAAGCAGCG-3′/Rev:5′CTCTTGGGAGCTTGGGACAG-3′. *ACTB*-Fw:5′-AGATGACCCAGATCATGTTTGAG-3′/Rev:5′-GTCACCGGAGTCCATCACG-3′. *GAPDH*-Fw:5′-CCTGTTCGACAGTCAGCCG-3′/Rev:5′-CGACCAAATCCGTTGACTCC-3′. *RPS15*-Fw:5′-CGACCAAATCCGTTGACTCC-3′/Rev:5′-CGGGCCGGCCATGCTTTACG-3′.

The RT-qPCR reaction began with heating at 95 °C for 10 min, followed by 45 cycles of denaturation at 95 °C for 30 s, annealing at the corresponding temperature for each gene (56–61 °C) for 20 s and extension at 72 °C for 30 s. Each assay included a negative control consisting of the absence of cDNA. Expression data were generated from two amplification reactions with samples and controls run in triplicate. Optical data obtained by RT-qPCR were analyzed using the MyiQ Single-Color Real-Time PCR Detection System Software v.1.0 (Bio-Rad, Hercules, CA, USA). Melt Curve analysis of each RT-qPCR assay and 1.5% agarose gel electrophoresis analysis of randomly selected samples were performed to confirm the specificity of the amplification products. Unless otherwise stated, the average expression of three different housekeeping genes (ACTB, GAPDH, and RPS15) was employed to normalize expression data using the Gene Expression Macro Software Version 1.1 (Bio-Rad Laboratories, Hercules, CA, USA), where the relative expression values were computed by the comparative Ct method [67,68].

For detection of mature miRNA, cDNA was prepared in a reverse transcription reaction using miScript HiSpec Buffer from the miScript II RT Kit (Qiagen Inc, Hilden, Germany). Oncogene miR-21 was detected by RT-qPCR. The primers used were Has-miR-21-5p (MS00009079, Qiagen Inc, Hilden, Germany) and RNU6-2 (MS00033740, Qiagen Inc, Hilden, Germany) as reference mature miRNA. The RT-qPCR assay was conducted under the following conditions: Stage 1: 15 min at 95 °C; Stage 2: 60 cycles of 15 s at 94 °C, 30 s at 55 °C, 30 s at 70 °C, 1 s at 72; and Stage 3: 5 s at 95 °C, 1 min at 65 °C. The real-time fluorescence intensity was monitored at each cycle of the third stage. Light Cycler^®^ 480 II Real-Time PCR System (Roche, Basilea, Switzerland) was used to perform the reaction.

### 4.7. RNA Sequencing

RNA sequencing study was performed in HEMn-LP melanocyte cell line transduced with RKIP shRNA Lentiviral Particles (sc-36430-V) or Control shRNA Lentiviral Particles (sc-108080) (Santa Cruz Biotechnology Inc., Dallas, TX, USA) After transfection, RKIP downregulation resulted in more than 80% and two independent HEMn-LP culture passages were used for RNA sequencing.

The RNA concentration measured by means of a Qubit 2.0 RNA assay Kit (Q32855, Invitrogen, Carlsbad, CA, USA) showed to be enough to start the experiment. The quality of all assayed RNAs (characterized by Agilent 2100 Bioanalyzer using an Agilent 6000 Nano Chip #5067-1511) was optimal. Sequencing libraries were prepared following “TruSeq Stranded mRNA Sample Preparation Guide (Part # 15031058 Rev. E)” with the corresponding kit [RS-122-2101 or RS-122-2102 (Set A or B, respectively) Illumina Inc., San Diego, CA, USA]. Briefly, starting from 400 ng of total RNA, mRNA was purified, fragmented, and primed for cDNA synthesis. cDNA first strand was synthesized with SuperScript-II Reverse Transcriptase (# 18064-014, Thermo Fisher Scientific, Waltham, MA, USA) for 10 min at 25 °C, 15 min at 42 °C, 15 min at 70 °C and pause at 4 °C. cDNA second strand was synthesized with Illumina reagents at 16 °C for 1 hour. Then, A-tailing and adaptor ligation were performed. Finally, enrichment of libraries was achieved by PCR (30 s at 98 °C; 15 cycles of 10 s at 98 °C, 30 s at 60 °C, 30 s at 72 °C; 5 min at 72 °C and pause at 4 °C). Afterwards, libraries were visualized on an Agilent 2100 Bioanalyzer using Agilent High Sensitivity DNA kit (# 5067-4626, Agilent Technologies, Santa Clara, CA, USA) and quantified using Qubit dsDNA HS DNA Kit (Q32854, Thermo Fisher Scientific, Waltham, MA, USA). Single-paired Sequencing was performed on the Illumina platform HiSeq2500 (resulting in 51 bp reads after discarding the final base). The images taken during the sequencing reactions were processed using Illumina’s sequencing control software for system control and base calling through an integrated primary analysis software called RTA (Real Time Analysis).

### 4.8. RNA Sequencing Data Analysis

The raw data have been deposited in NCBI Sequence Read Archive (SRA) through the Gene Expression Omnibus (GEO, accession number GSE151585), and a supervised similarity analysis was performed. Reads were initially mapped to the reference genome from the UCSC genome browser (http://genome.ucsc.edu/; Human Dec. 2013 GRCh38/hg38 assembly), using Hisat2. Reads were assembled into transcripts using StringTie. Initial quality analysis (Appendix A) showed a strong congruence of the biological replicates: Spearman correlation values for all replicates were r = 0.81 demonstrating the reliability of the data produced and illustrating the consistency of the transcriptional changes within each condition.

Differentially expressed genes were identified utilizing edgeR package in R-studio, by implementing a negative binomial distribution for the statistical significance. Normalization was performed using the trimmed mean of M values (TMM) method. Reads per kilobase of exon model per million mapped reads (RPKM) for each gene were transformed to log2-fold changes using the R 3.1.0 package (https://www.r-project.org/; R Core Team, 2014). The analysis of variance (*p* < 0.05) and false discovery rate (FDR < 0.05) tests were performed using the R program (version 3.1.0) to select genes exhibiting significantly different expression patterns. The upregulated and downregulated differential expression gene sets were generated by extracting the 500 genes with the respectively highest and lowest values from the gene expression signature. Enrichment results were generated by analyzing the upregulated and downregulated gene sets using BioJupies, which is freely available at https://amp.pharm.mssm.edu/biojupies/ and EnrichR, which is freely available at https://amp.pharm.mssm.edu/Enrichr/, using the gene set upload API. The following analyses were carried out: Gene Ontology Analysis, Enriched Pathways Analysis, Enriched Transcription Factor Analysis, Enriched microRNA Analysis.

### 4.9. Statistical Analysis

All experiments were performed at least three times. Unless otherwise stated, calculations were performed with IBM SPSS Statistics for Windows, version 26 (SPSS Inc., Chicago, IL, USA) using Fisher’s exact test. Mean values and standard deviation (SD) were calculated and indicated. *p* values below 0.05 were regarded as significant and marked by an asterisk (*).

## 5. Conclusions


*What does this study add?*
✓RKIP discerns between benign and malign melanocytic lesions by immunohistochemistry.✓RKIP regulates NANOG transcription factor, inferring in the invasion capacity of melanoma cells via regulation of EMT.



*What is the translational message?*
✓The laboratory study demonstrated that RKIP could be a useful marker for melanoma diagnosis.✓Functional analyses indicate that RKIP plays an important role in the maintenance of cellular characteristics related to a differentiated state. Therefore, loss of RKIP would drive a dedifferentiation process, which often correlates with resistance to chemotherapy. So, upregulation of RKIP could be a strategy to analyze for the management of melanoma patients who present with resistance to currently available treatments.


## Figures and Tables

**Figure 1 cancers-12-01451-f001:**
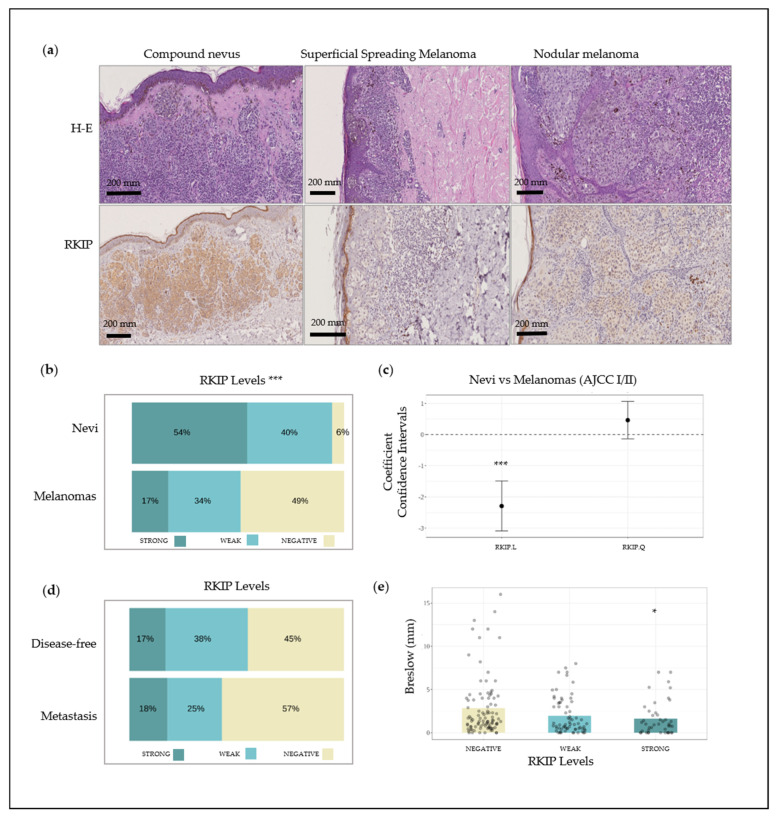
Raf Kinase Inhibitor Protein (RKIP) expression in FFPE biopsies from patients. (**a**) Lower section of picture: IHC analysis of RKIP in normal melanocytes from a compound nevus, superficial spreading melanoma and nodular melanoma. Upper section of picture: Hematoxylin-Eosin staining (H-E); (**b**) RKIP staining distribution on nevus and melanoma tissue; (**c**) Coefficient confidence intervals for RKIP protein expression between nevus and melanoma samples; (**d**) RKIP staining distribution on histological sections of melanoma in stages I and II from patients who remained disease-free during follow-up versus patients who developed metastasis; (**e**) Kruskal–Wallis one-way analysis of Breslow index with respect to RKIP expression. * q < 0.05, *** q < 0.001.

**Figure 2 cancers-12-01451-f002:**
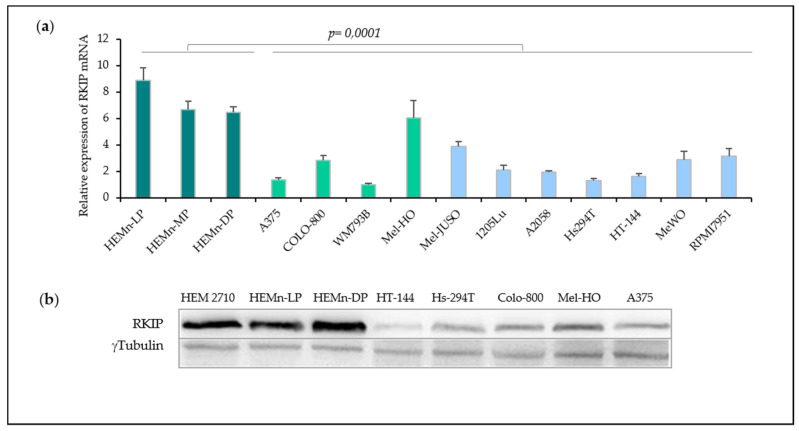
RKIP expression in cell culture of primary melanocytes and melanomas. (**a**) RKIP mRNA expression in primary melanocytes and melanoma cell lines. RNA from three human melanocytes (dark green) lightly (HEMn-LP), moderately (HEMn-MP), and darkly (HEMn-DP) pigmented neonatal foreskin lines together with four primary melanomas cell lines (light green) and seven metastatic melanomas cell lines (light blue) were analyzed by RT-qPCR. (**b**) RKIP protein expression in the three melanocytes cell lines and five melanoma cell lines assessed by Western Blot. Tubulin expression was used as loading control. Figure is representative of three independent experiments.

**Figure 3 cancers-12-01451-f003:**
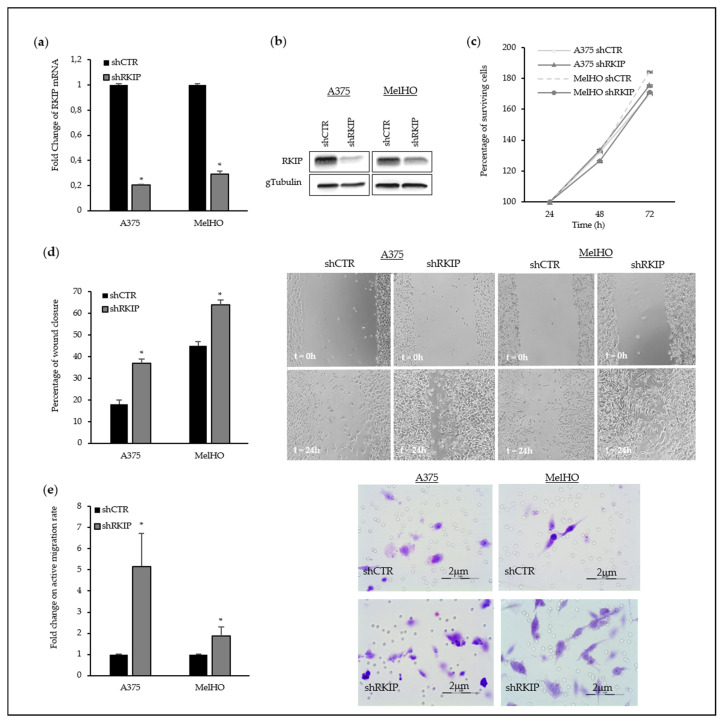
Modulation of RKIP expression in primary melanoma cell lines. (**a**) RKIP mRNA levels in RKIP-downregulated A375 and MelHO primary melanoma cell lines. A375 and MelHO cells were transduced with RKIP shRNA Lentiviral Particles or Control shRNA Lentiviral Particles following the manufacturer´s instruction. Two days after infection, the cells were selected with Puromycin to get stable cell lines; (**b**) Western Blot assay showed the RKIP-downregulation in A375 and MelHO melanoma cells; (**c**) Proliferation rate in A375 and MelHO primary melanomas after RKIP downregulation. The viability of control melanoma cells (with an empty vector) and stable RKIP transfected clones were subjected to XTT assays for 24, 48, and 72 h. Results of each experiment are expressed related to the values obtained for the transfection control. Data is given as a mean ± SD of at least three experiments of different transfection; (**d**) Fold change on wound healing rate in A375 and MelHO primary melanoma after RKIP downregulation; (**e**) Fold change on active migration rate in presence of collagen in primary melanoma after RKIP downregulation. The histograms in (**d**) and (**e**) show the average of three independent assays with six replicates per assay and representative pictures have been included. * *p*-value < 0.05.

**Figure 4 cancers-12-01451-f004:**
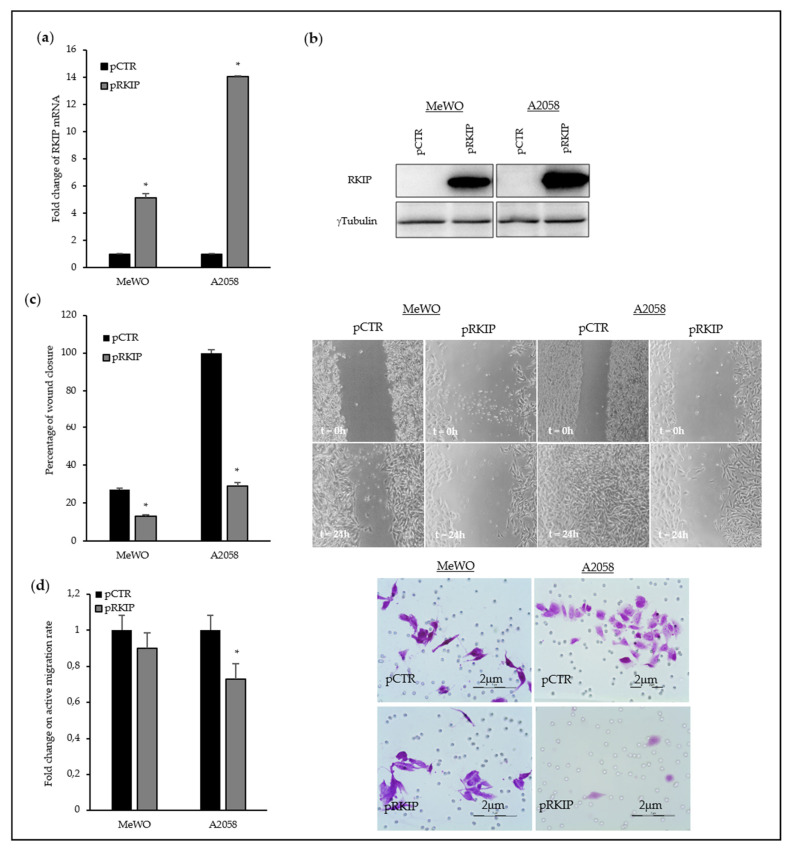
Modulation of RKIP expression in MeWO and A2058 metastatic melanoma cell lines. (**a**) RKIP mRNA levels in RKIP-upregulated MeWO and A2058 metastatic melanoma cell lines. A2058 and MeWO cell lines were transfected with overexpressing plasmid for RKIP (pRKIP) or empty plasmid (pCTR) using Lipofectamine 2000 as transfection agent. All of the transfection experiments were performed with 500 ng of DNA. (**b**) Western Blot assay showed the RKIP-upregulation in MeWO and A2058 melanoma cells; (**c**) Fold change on wound healing rate in MeWO and A2058 metastatic melanoma after RKIP upregulation. The experimental assays were performed at least after 24 h of RKIP transfection; (**d**) Fold change on active migration rate in presence of collagen in metastatic melanoma after RKIP upregulation. The histograms in (**c**) and (**d**) show the average of three independent assays with six replicates per assay and representative pictures have been included. * *p*-value < 0.05.

**Figure 5 cancers-12-01451-f005:**
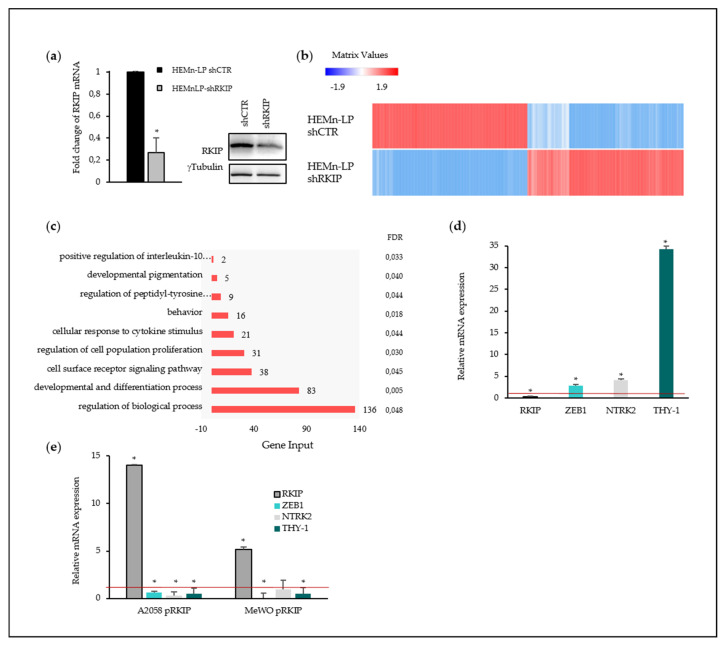
RNA Sequencing data and analysis. (**a**) RKIP mRNA and protein levels in RKIP-downregulated HEMn-LP. Normal melanocyte cell line HeMn-LP was transduced with RKIP shRNA Lentiviral Particles or Control shRNA Lentiviral Particles following the manufacturer´s instruction. Two days after infection, the cells were selected with Puromycin to get stable cell lines. The RKIP downregulation were validated by RT-qPCR and Western Blot; (**b**) Clustergram analysis showing differential expression genes data set comparing controls versus RKIP knockdown HEMn-LP cells; (**c**) Every row of the figure represents one enriched process after RKIP downregulation with an FDR cutoff of 0.05; (**d**) Relative expression of three selected genes for RNASeq results validation in shRKIP HEMn-LP; (**e**) Relative expression of three selected genes for RNASeq results validation in metastatic melanoma cells after RKIP upregulation. (**d**,**e**) *ACTB* was used as a housekeeping gene for relative quantification. The average of three independent assays have been shown. The red line highlights the control normalized expression level. * *p*-value < 0.05.

**Figure 6 cancers-12-01451-f006:**
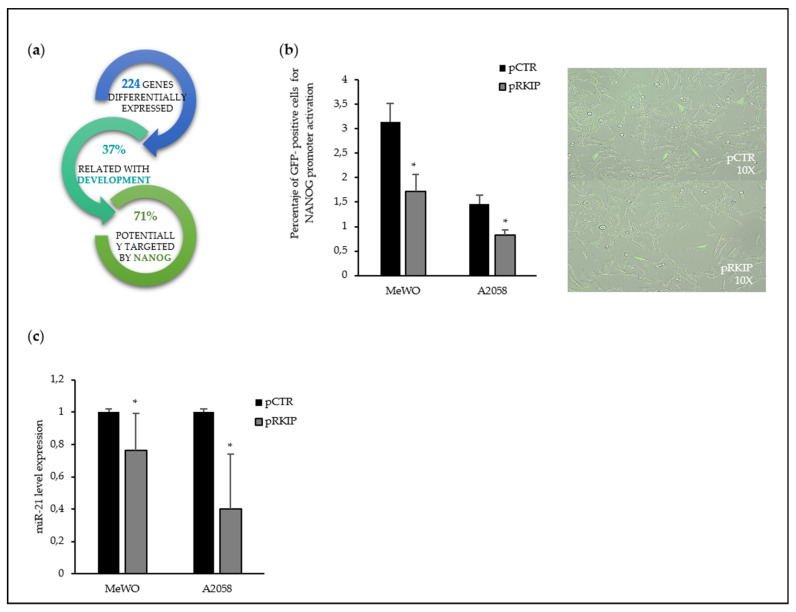
RKIP as a key regulator of NANOG expression in melanoma. (**a**) Overview of differential expressed genes after RKIP silencing related to development and NANOG transcription factor; (**b**) Co-transfection of RKIP overexpressing plasmid and GFP-NANOG promoter construct in MeWO and A2058 metastatic melanoma cell lines. It is shown the positive cells for GFP emission from three independent assays and a representative picture of one of the A2058 cells assay; (**c**) MicroRNA-21 level determination by RT-qPCR in RKIP upregulated melanoma cell lines RNU6-2 was used as reference mature miRNA. Average of three independent assays per cell line. * *p*-value < 0.05.

**Table 1 cancers-12-01451-t001:** Clinical and pathological data from nevus and melanoma patients.

Characteristics	N (%)
NEVUS	75
Age at diagnosis (years, range)	56 (24–78)
Sex	
Male	28 (37)
Female	47 (63)
MELANOMAS	239
Age at diagnosis (years)	57 (23–87)
Sex	
Male	131 (55)
Female	108 (45)
Localization	
Head and neck	43(18)
Trunk	74(31)
Upper limb	24 (01)
Lower limb	69 (29)
Acral	21 (9)
Others	5 (2)
ND	3 (1)
Histological subtype	
SSM	102 (43)
NM	53(22)
ALM	21(9)
LMM	9(4)
LM	3 (1)
Others	12 (5)
ND	39 (16)
AJCC stage at diagnosis	
In situ	34 (14)
IA	46 (19)
IB	54 (23)
IIA	32 (13)
IIB	15 (6)
IIC	24 (10)
IIIA	9 (4)
IIIB	11 (5)
IIIC	6 (3)
IV	8 (3)
Disease progression	
Disease-free	147 (62)
Metastasis	92 (38)

Abbreviations: SSM, Superficial Spreading Melanoma; NM, Nodular Melanoma; ALM, Acral Lentiginous Melanoma; LMM, Lentigo Maligna Melanoma; LM, Lentigo Maligna.

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
