# Peer review of "RKIP Regulates Differentiation-Related Features in Melanocytic Cells"

_cancers, 2020, doi:10.3390/cancers12061451_

Round 1

Reviewer 1 Report

The authors have created a well written manuscript describing the importance of using RKIP as a diagnostic marker for melanoma.  The data is well presented and the conclusions and limitations are clearly described.

Comments:

  1. In the Results section, the verb selection in the two first sentences, “to deepen on the mechanism”, is awkward. “Melanocytes and several malignant melanoma cell lines were selected in order to deepen on the mechanisms underlying observed differences in human samples” – maybe; Melanocytes and several malignant melanoma cell lines were selected in order to further decipher the mechanisms underlying…..
  2. Were all three house-keeping genes used for each qPCR reaction? If not, the figure legend should indicate which house-keeping gene was used.
  3. Are there images missing for figure 3E? The histograms in (d) and (e) shows the average and representative pictures of three independent assays with six replicates per assay.
  4. What are the p values for bar graphs in figure 3d and 3e? There is no indication in the figure legend. P values or a description of * should be added to figure legends for bar graphs.
  5. Line 182, why is there a “?” after RKIP?
  6. Line 302, should be “associated with metastatic uveal melanoma”
  7. Line 360, “RKIP expression on melanocytes resulted on the”..- “on” should be “in”

Author Response

First of all, we would like to thank the referee for the positive evaluation of our work.

Comments:

In the Results section, the verb selection in the two first sentences, “to deepen on the mechanism”, is awkward. “Melanocytes and several malignant melanoma cell lines were selected in order to deepen on the mechanisms underlying observed differences in human samples” – maybe; Melanocytes and several malignant melanoma cell lines were selected in order to further decipher the mechanisms underlying….. 

Change has been done in Section 2.2, page 4, line 131.Were all three house-keeping genes used for each qPCR reaction? If not, the figure legend should indicate which house-keeping gene was used. Yes; unless otherwise stated, three house-keeping genes were employed to relativize each reaction. This information has been included in Section 4.6, page 17, line 566 that now reads as follows:“Unless otherwise stated, the average expression of three different housekeeping genes (ACTB, GAPDH, and RPS15) was employed to normalize expression data using the Gene Expression Macro Software Version 1.1 (Bio-Rad Laboratories, Hercules, CA, USA), where the relative expression values were computed by the comparative Ct method [67,68]”

In addition, we have added specific information for Figure 6 (page 10, line 288): “RNU6-2 was used as reference mature miRNA. Average of three independent assays per cell line. * p-value < 0,05.”

Are there images missing for figure 3E? The histograms in (d) and (e) shows the average and representative pictures of three independent assays with six replicates per assay. Pictures for insert assays have been added to the Figure 3 (page 6) and Figure 4 (page 7). Legends have been also corrected.

What are the p values for bar graphs in figure 3d and 3e? There is no indication in the figure legend. P values or a description of * should be added to figure legends for bar graphs. We thank the referee for highlighting this mistake. P-value of bar graphs as well as corresponding * have added in each figure legend: Figure 3 (page 6), Figure 4 (page 8), Figure 5 (page 8-9), Figure 6 (page 10).

Line 182, why is there a “?” after RKIP?Misspelling; correction can be found in page 6, line 183.

Line 302, should be “associated with metastatic uveal melanoma”It has been corrected accordingly (page 11, line 305). 

Line 360, “RKIP expression on melanocytes resulted on the”...- “on” should be “in” It has been corrected accordingly (page 12, line 366). 

Reviewer 2 Report

Dear Authors;

This work is really interesting especially because of the use of different methods to investigate the role of RKIP in melanoma, starting from an easy diagnostic technique as immunohistochemistry. Immunohistochemical expression of RKIP should be better described in terms of the percentage of positive cells, signal intensity, and localization of immunohistochemical signal. Moreover, 51% of tested melanomas presented positive staining (17% strong and 34% weak) and 80% of in situ melanomas showed strong positivity, but images of in situ melanomas are not presented. The association between the Breslow index and levels of RKIP expression is the strength of the work and should be better explained and emphasized. Much more images of RKIP immunohistochemical expression should be exhibited in particular examples of STRONG, WEAK, and NEGATIVE expression, to better understand the differences between the established categories. RKIP could be considered a supporting marker to discriminate between nevi and melanoma but it discerns between benign and malign melanocytic lesions when strongly expressed only, in my opinion. Figure 2 shows that the Mel-HO cell line (primary melanoma cell line) displays high levels of RKIP mRNA, similar to melanocyte cell line, and this discrepant event should be better explained.

The relationship between RKIP and NANOG expression is really interesting and opens up the possibility of important future researches.

Author Response

First of all, we would like to thank the referee for the positive evaluation of our work. As for the immunohistochemistry (IHC) we employed a manual semi-quantitative scoring system based on the analysis of samples by two evaluators independently. RKIP staining was homogenous among melanocytic cells from each sample and therefore, we did not consider the percentage of positive cells. Categories for signal intensity were established according to the pattern included now as Supplementary Figure A1. Of note, RKIP signal was cytoplasmic.

This information has been included in Section 4.2 (page 14, line 452) that now reads as follows:

Analysis of samples was conducted following a manual semi-quantitative method; categories were determined based on staining intensities (negative (0), weak (1) and strong (2)) as shown in Supplementary Figure A1. Samples displayed a uniform intra-specimen staining and RKIP marker presented high affinity to melanocytic and melanoma cells (no staining observed in surrounding cells). Due to the color similarities among melanin, macrophage pigment and chromogen signal, any misinterpretation was avoid by the analysis of a sequential slide stained with hematoxylin-eosin.”

Also has been modified section 2 (page 3, line 100) and now reads as follows:

“Interestingly enough, in situ melanomas, characterized by an excellent prognosis upon surgical removal, exhibited a strong positive RKIP expression in almost 80% of cases (Figure A1). RKIP staining displayed minimum intrasample variation and a cytoplasmic localization.”

Moreover, 51% of tested melanomas presented positive staining (17% strong and 34% weak) and 80% of in situ melanomas showed strong positivity, but images of in situ melanomas are not presented.

This information has been included as a new figure named as “Figure A1”.

The association between the Breslow index and levels of RKIP expression is the strength of the work and should be better explained and emphasized.

As suggested by the referee, we have extended the explanation regarding the association between the low Breslow index and high expression of RKIP. This information has been included in Section 3 (Discussion), starting on page 12, line 391 and now reads as follows:

“In addition to the maintenance of the stemness, NANOG has been also implicated in the EMT [19,26] by regulating the expression of ZEB1 and THY-1 among other genes [28]. In fact, EMT and development of stemness properties are often closely related processes [58]. As previously mentioned, these two genes are among those with deregulated expression in our RNA-seq study. ZEB1 is one of the major activators of the EMT program and increasing evidence places ZEB1 also as an important regulator of differentiation, proliferation, DNA damage response and cell survival [59]. Interestingly enough, ZEB1 is among the transcription factors driving the early hybrid EMT state and hybrid EMT states (i.e. states with intermediate characteristics among fully epithelial and fully mesenchymal cells) have been linked to collective cells migration and highest metastatic potential [58]. This result, together with the observed modulation of the cellular migration capacity driven by RKIP, are in line with the rapid RKIP diminution observed on malignant lesions, even at early stages, as well as the association among low Breslow index and presence of RKIP. In fact, capacity of a tumor to deepen on the skin requires the acquisition of characteristics as those blocked by RKIP.”

Much more images of RKIP immunohistochemical expression should be exhibited in particular examples of STRONG, WEAK, and NEGATIVE expression, to better understand the differences between the established categories.

We thank the referee for highlighting this point. We have incorporated this information in the new figure “Figure A1”. This new figure has been conceived to summarize and answer to some of the aspects mentioned by the referee.

RKIP could be considered a supporting marker to discriminate between nevi and melanoma but it discerns between benign and malign melanocytic lesions when strongly expressed only, in my opinion.

The referee is right although we would like to add that negative RKIP staining is also very informative. Lesions with strong RKIP staining are likely to be benign while those with negative staining are generally malignant melanomas. Furthermore, it is precisely this remarkable difference in strong and negative staining between nevi and melanoma what the result for the linear polynomial contrast in the multivariate model of Figure 1c is reflecting. On the other hand, the group known as “Weak staining” was more heterogeneous which would be the reason underlying the lack of difference among nevi and melanomas in this category. This may have been a limitation of our study that we would like to consider for the future.

Figure 2 shows that the Mel-HO cell line (primary melanoma cell line) displays high levels of RKIP mRNA, similar to melanocyte cell line, and this discrepant event should be better explained.

Regarding this point, we can only speculate based on information obtained from literature. The high mRNA level (when compared with other melanoma cell lines) but an average protein content (compared with other melanoma cells lines) suggests the involvement of a post-transcriptional mechanism possibly by a microRNA. In this regard, a review by Zaravinos et al. (2018) mentions several microRNAs able to downregulate RKIP protein expression (e.g. miR-224, miR-543). Consequently, we have slightly modified the manuscript in Section 2.2, page 4, line 135:

The relative quantification of the mRNA levels evidenced a generalized reduction of RKIP expression in melanoma cell lines (p-value=0.0001) compared to primary normal melanocytes (Figure 2a).

In addition, we have discussed this finding in Section 3, page 11, line 340 that now reads as follows:

In accordance with our histopathological results and previous published studies [14,36], we found that both RKIP mRNA and protein expression were significantly lower in melanoma cells lines than in primary cultures of melanocytes with the exception of the Mel-HO cell line; this cell line exhibited RKIP mRNA level similar to that observed on melanocytes but a reduced protein content that suggests the involvement of a post-transcriptional mechanism limiting translation. Of note, RKIP has been described as a target for several microRNAs able to regulate cellular protein level [2].

The relationship between RKIP and NANOG expression is really interesting and opens up the possibility of important future researches.

We agree with the referee.
